# Translocated populations are genetically similar to natural populations and populations resulting from natural colonizations

José F. Meléndez-Cal-y-Mayor[1]*, Jasmin Winkler[1], Ramon Müller[1], Beatrice Lüscher[2], Janine Bolliger[3], Arpat Ozgul[1], Benedikt R. Schmidt[1,4]

**1** Institute of Evolutionary Biology and Environmental Science, University of Zurich, Zürich, Switzerland, **2** Info Fauna KARCH Regional Representation, Canton of Bern, Amphibians, Münsingen, Switzerland, **3** WSL, Swiss Federal Research Institute for Forest, Snow and Landscape Research, Birmensdorf, Switzerland, **4** Info Fauna Karch, Neuchâtel, Switzerland

* jose.melendezcalymayor@gmail.com

## Abstract

Genetic diversity and structure are rarely assessed in populations established through conservation translocation. Here, we analysed the genetic structure and diversity of populations of an endangered pond-breeding amphibian, the common midwife toad, *Alytes obstetricans*, by comparing translocated populations against two types of populations: (i) populations which have recently colonized newly created ponds and (ii) natural populations which have been known to be present for a long time. Bottleneck events and dispersal patterns were analysed to describe the outcome of the translocations. In addition, we simulated trajectories of genetic diversity ($H_e$) of populations over time. The genetic diversity of natural colonized and translocated populations was similar to that of natural populations. However, there were signatures of genetic bottleneck events in three colonized populations and in a natural population. Simulations of genetic diversity over time showed that number and frequency of dispersers and population size are important parameters determining genetic diversity in the populations in the future. We conclude that, translocated, natural and colonized populations are genetically comparable, indicating that translocations can serve as an effective tool in conservation efforts. However, constructing ponds and waiting for natural colonization is also known to work well.

## Introduction

Conservation biology has two main goals. The first goal is to understand how anthropogenic stressors lead to the loss of biodiversity. The second equally important goal is to work on solutions that mitigate the negative effects of anthropogenic stressors on populations [1]. To bend the curve of biodiversity loss, solutions are needed to restore habitats and populations or to slow or halt the decline of populations to allow

**Data availability statement:** The Zenodo DOI is already available: https://doi.org/10.5281/zenodo.15052694.

**Funding:** The author José F. Meléndez-Cal-y-Mayor received a scholarship from the Consejo Nacional de Humanidades, Ciencias y Tecnologías (CONAHCYT; Award Number: 438370). The author Benedikt R. Schmidt received an award from the University of Zurich. The funders had no role in study design, data collection and analysis, decision to publish, or preparation of the manuscript.

**Competing interests:** The authors have declared that no competing interests exist.

the recovery of declining species [2–4]. Once solutions are available and implemented by conservation practitioners, it is necessary to assess their effectiveness [5–7]. Solutions such as habitat creation or restoration allow target species to colonize those habitats [6,8,9]. However, in some cases, natural colonizations may not be possible or take a long time [10]. In addition, colonized populations may have low genetic diversity due to the founder effect (i.e., bottleneck), caused by the small number of founder individuals and the small number of founder populations [11,12]. Conservation translocations may therefore have a role in wildlife conservation because one may select a translocation strategy that maximizes genetic diversity [13].

Translocations are commonly used in conservation practice [14–16]. A conservation translocation is the intentional release of organisms at a site with the goal to establish, reestablish, or augment a population [14,17]. The conservation benefit is achieved, and a translocation is viewed as successful, when it has led to a viable population [18–20]. However, the success of a viable translocated population may be jeopardized if translocated individuals have little genetic diversity [21,22]. Evidence suggests that some translocated populations have reduced levels of genetic diversity when compared to long established populations [23–26]. There are many reasons for reduced genetic diversity in translocated populations. For instance, the number of translocated founders, admixture, genetic drift, and bottlenecks may determine the genetic diversity of the translocated population [27,28]. A high number of translocated individuals and multiple source populations reduce the likelihood of bottlenecks, inbreeding and genetic drift [29], and should therefore lead to translocated populations with levels of genetic diversity similar to natural population. Inbreeding and reduced levels of genetic diversity can affect population viability [30–32]. Because genetic diversity can affect long-term population viability [33–35], it is important to compare the genetic diversity of translocated populations to long-established natural populations and to populations which have recently colonized previously empty patches. This is also important because genetic diversity is often neglected in conservation practice [36].

Amphibians are declining globally [37–40]. To aid the recovery of amphibians, translocations are commonly used [4,29,41,42]. In this study, we evaluated translocations using microsatellites to document the genetic diversity and structure of populations of the common midwife toad, *Alytes obstetricans*. We used data from two conservation programs which had the aim to increase the number of populations in the Swiss Emmental and in the Lucerne regions (genetic rescue was not an aim; [6,43–46]) to compare three population types in two regions within Switzerland: natural populations, translocated populations and populations which resulted from natural colonizations (the local amphibian conservation officer Beatrice Lüscher made sure that no translocations took place in these naturally colonised populations) after the creation of new ponds. Ponds in the amphibian populations were surveyed decades ago; therefore, we can be confident that natural populations were present already decades ago and persisted (for reviews of the surveys, see [43] (Lucerne) and [44] (Emmental). This allows us to compare natural versus translocated populations and natural versus colonized populations. We do not expect the effect of translocations

and natural colonizations on the genetic structure of the new populations to be necessarily different. While conservationists worry about the genetic consequences of translocations, this is usually not the case for natural colonizations. We believe that a comparison of the two types of new population may be useful for conservation science and practice. The lessons learned from this study can improve future conservation actions because the genetic consequences of translocations are better understood and compared to demographic processes such as colonization within a metapopulation.

## Materials and methods

### Research permits and ethical considerations

Permission to capture specimens of *Alytes obstetricans* were granted by the nature conservation authorities of the Swiss cantons Lucerne and Bern (to Benedikt Schmidt). Animal welfare permits to collect tissue samples were granted by the animal welfare agencies of the Swiss cantons Zurich and Bern (ZH66/2014(an intercantonal permit), BE17/15). All procedures have been carried out in accordance with relevant guidelines and regulations and, where applicable, reported in accordance with ARRIVE guidelines. To minimise suffering, the tadpoles were anaesthetised using ethyl 3-aminobenzoate methanesulfonate (Sigma-Aldrich, Buchs, Switzerland, code MS-222) before tissue sampling. Tadpoles were subsequently released. No animals were sacrificed.

### Study species and sampling sites

The midwife toad (*Alytes obstetricans*) is classified as "least concern" on the global IUCN Red List. However, as in many other European countries [47], it is endangered in Switzerland due to ongoing population declines [48–50].

The sampling sites are located in central Switzerland around Lucerne and Emmental (Fig 1). Genetic samples from all known populations of *A. obstetricans* in Emmental and Lucerne (n = 33) were collected between April and July 2015. Twenty-one sites were in Emmental of which twelve were natural populations and nine were naturally colonized (Table 1).

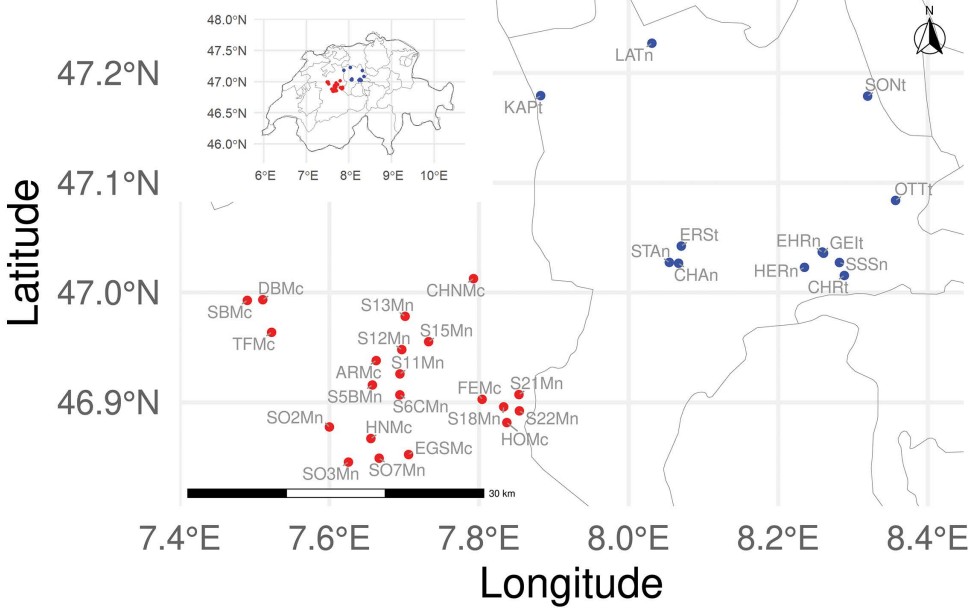

**Fig 1. Sampled populations located in central Switzerland.** The red dots show the populations sampled in the Emmental (natural and colonized populations) and the blue dots show the populations sampled in Lucerne (natural and translocated populations). The lowercase letters at the end of the names explains the population type (n = natural, t = translocated and c = colonized). The elevations of the study sites range from 528 to 1127 m.a.s.l. The inset map shows where in Switzerland the sites are located.

**Table 1. Gene diversity indicators for natural, colonized and translocated populations at the study sites. The table lists the area, the population type, the population, sample size and the estimated genetic metrics. The confidence intervals (95%; values in brackets) for effective population size ($N_e$) were obtained using the jackknife resampling method.**

| Region | Population type | Population | Sample size (n) | Allelic richness ($A_R$) | Observed Heterozygosity ($H_o$) | Expected heterozygosity ($H_e$) | Fixation index ($F_{ST}$) | Private alleles ($P_a$) | Inbreeding coefficient ($F_{is}$)/Median values | Effective population size ($N_e$) |
|---|---|---|---|---|---|---|---|---|---|---|
| Emmental | Colonized | ARM | 15 | 1.45 | 0.184226 | 0.163407 | 0.364 | 2 | −0.133/-0.167 | Infinite (10.9-Infinite) |
| | | CHNM | 8 | 1.72 | 0.265625 | 0.258333 | 0.305 | 0 | −0.03/0.000 | Infinite (3.4-Infinite) |
| | | DBM | 10 | 1.25 | 0.150000 | 0.111184 | 0.439 | 0 | −0.376/-0.378 | 11.8 (0.5-Infinite) |
| | | EGSM | 20 | 1.93 | 0.298191 | 0.303590 | 0.291 | 3 | 0.018/0.059 | 86.8 (18.7-Infinite) |
| | | FEM | 17 | 1.94 | 0.330882 | 0.326822 | 0.231 | 6 | −0.013/-0.016 | 12.4 (2.1-Infinite) |
| | | HNM | 20 | 1.97 | 0.302303 | 0.326531 | 0.247 | 1 | 0.076/0.036 | Infinite (31.1-Infinite) |
| | | HOM | 20 | 1.85 | 0.275822 | 0.274193 | 0.263 | 4 | −0.006/0.028 | 2.7 (1.5-9.1) |
| | | SBM | 20 | 1.38 | 0.121875 | 0.112420 | 0.446 | 1 | −0.087/-0.042 | 8.1 (1.7-Infinite) |
| | | TFM | 15 | 1.76 | 0.236310 | 0.243183 | 0.278 | 0 | 0.029/0.029 | 1.8 (0.6-17.0) |
| | Natural | SO2M | 6 | 2.00 | 0.229167 | 0.223485 | 0.298 | 1 | −0.028/0.000 | Infinite (3.5-Infinite) |
| | | S13M. | 20 | 2.21 | 0.308224 | 0.312276 | 0.225 | 6 | 0.013/-0.042 | Infinite (23.9-Infinite) |
| | | S15M. | 20 | 2.03 | 0.290625 | 0.288301 | 0.258 | 0 | −0.008/-0.006 | 50.3 (12.9-Infinite) |
| | | S18M. | 12 | 1.74 | 0.234375 | 0.229846 | 0.269 | 0 | −0.021/0.000 | 20.4 (4.3-Infinite) |
| | | S21M. | 20 | 1.74 | 0.275000 | 0.239423 | 0.283 | 0 | −0.153/-0.148 | 47.1 (5.9-Infinite) |
| | | S22M. | 7 | 1.95 | 0.339286 | 0.287088 | 0.246 | 0 | −0.2/-0.163 | Infinite (4.7-Infinite) |
| | | SO3M | 20 | 1.55 | 0.218750 | 0.189984 | 0.327 | 3 | −0.156/-0.137 | 49.5 (6.1-Infinite) |
| | | SO7M | 20 | 1.60 | 0.225000 | 0.198157 | 0.376 | 0 | −0.14/-0.101 | 8.4 (2.1-73.3) |
| | | S11M. | 14 | 1.63 | 0.252232 | 0.237747 | 0.307 | 0 | −0.064/-0.030 | 5.5 (1.0-Infinite) |
| | | S12M. | 20 | 1.76 | 0.270703 | 0.257939 | 0.282 | 0 | −0.051/-0.027 | 7.4 (1.7-214.0) |
| | | S5BM. | 20 | 1.74 | 0.233717 | 0.223537 | 0.306 | 1 | −0.047/-0.071 | 6.1 (1.7-41.7) |
| | | S6CM. | 20 | 2.03 | 0.322606 | 0.291544 | 0.239 | 0 | −0.11/-0.065 | 8.7 (2.6-36.8) |
| Lucerne | Translocated | KAP | 30 | 1.87 | 0.245833 | 0.232839 | 0.312 | 10 | −0.057/-0.084 | 171.4 (20.7-Infinite) |
| | | GEI | 21 | 1.79 | 0.254315 | 0.257341 | 0.367 | 2 | 0.012/-0.060 | 16.2 (6.2-87.4) |
| | | OTT | 19 | 1.65 | 0.240439 | 0.240995 | 0.378 | 1 | 0.002/0.016 | 2.3 (1.3-8.0) |
| | | CHR | 30 | 2.13 | 0.295833 | 0.303884 | 0.256 | 1 | 0.027/0.008 | 25.9 (12.9-82.2) |
| | | ERS | 30 | 2.19 | 0.358333 | 0.352295 | 0.267 | 10 | −0.017/-0.039 | 14.8 (6.7-40.3) |
| | | SON | 30 | 2.48 | 0.400431 | 0.378446 | 0.206 | 15 | −0.059/-0.073 | 4.9 (2.6-8.9) |
| | Natural | EHR | 30 | 2.10 | 0.304167 | 0.292373 | 0.270 | 0 | −0.041/-0.057 | Infinite (34.4-Infinite) |
| | | CHA | 30 | 1.81 | 0.261422 | 0.232035 | 0.346 | 0 | −0.129/-0.122 | 64.4 (14.1-Infinite) |
| | | HER | 30 | 1.86 | 0.249138 | 0.238425 | 0.303 | 1 | −0.046/-0.047 | 1063.4 (29.5-Infinite) |
| | | LAT | 20 | 1.93 | 0.337500 | 0.306170 | 0.366 | 28 | −0.105/-0.082 | 3.2 (1.7-13.7) |
| | | SSS | 30 | 1.97 | 0.308333 | 0.310134 | 0.257 | 5 | 0.006/-0.012 | 11.5 (6.1-22.2) |
| | | STA | 30 | 2.06 | 0.281250 | 0.278143 | 0.275 | 16 | −0.011/-0.031 | Infinite (37.9-Infinite) |

In Lucerne, there were twelve populations. Six were natural populations (four out of six EHRL, CHAL, SSSL and STAL were used as donor populations) and six were translocated populations (Table 1). In Emmental we sampled 199 and 145 individuals from natural and colonized populations, respectively. In Lucerne we sampled 170 and 160 individuals from natural and translocated populations, respectively. More information on the number of individuals sampled per population is provided in Table 1. We disinfected field equipment and boots with Virkon S each time we collected samples from different sites to prevent the spread of disease during field work [51].

## Genetic data

*A. obstetricans* tadpoles were collected by dip-netting. For every population a maximum of 30 tadpoles were caught. Three mm of the tail tip were cut off with a scalpel blade. This is a method approved by the Swiss federal conservation and animal welfare agencies [52]. Tissue samples were put in 94% ethanol and stored in a freezer (−20.5 °C ± 0.8 °C) prior to laboratory analysis. We extracted DNA from the tail of a tadpole.

For DNA extraction, we used the BioSprint 96 DNA Blood Kit (Qiagen, Hombrechtikon, Switzerland, code 940054) following its protocols. The genetic data was obtained by microsatellite markers. The markers were previously used to analyse the genetic structure of Swiss populations of *Alytes obstetricans* [46]. Polymerase chain reaction (PCR) was performed with fluorescent labelled primers. Twelve markers were developed by Tobler et al. (2013) and four new markers were developed by ecogenics GmbH (Zurich, Switzerland; S1 Table). Linkage disequilibrium (LD) and deviation from Hardy-Weinberg equilibrium (HWE) were calculated with the R package adegenet (v2.1.7; [53,54]). Probable presence of null alleles was checked with FreeNA program [55]. The percentage of missing alleles in both datasets, Emmental and Lucerne, was calculated with the R package poppr (v2.9.5; [56,57]) using the functions *read.genalex* () and *info_table* (). On average, Emmental has 0.8% of missing alleles (S1 Fig) and Lucerne has 0.78% (S2 Fig).

## Genetic diversity and bottleneck

Inference of the genetic parameters rarefied allelic richness ($A_r$), observed and expected heterozygosity ($H_o$ and $H_e$, respectively), and fixation index ($F_{st}$; between populations within each region) was calculated using FSTAT (v2.9.4; [58]) and GDA (v1.1; [59]). The average of all the pairwise $F_{st}$ values per population was also used for analysis. The number of private alleles ($P_a$) per population was calculated using the *private_alleles* function in the package poppr (v2.9.5; [56,57]). The inbreeding coefficient ($F_{is}$) was estimated using FSTAT (v2.9.4; [58]). We used a one-sample t-test to assess the alternative hypothesis that the median $F_{is}$ values are not equal to zero with the function *t.test* () in R (v4.3.1; [60]). Effective population size ($N_e$) was estimated using the Linkage Disequilibrium (LD) method [61–63], as implemented in $N_e$Estimator (v2.1; [64]). The $N_e$ values for the populations in Emmental (ARM, CHNM, HNM, SO2M, S13M, S22M) and in Lucerne (EHR and STA) were excluded from further analysis because they were infinite. As explained by [63], infinite estimates of $N_e$ can arise due to sampling variation when the number of individuals in the sample is small. We believe that this interpretation of infinite $N_e$ values is more likely than genuinely high $N_e$ (which we believe are unlikely). Although the LD method yielded infinite $N_e$ estimates, the lower limits of the 95% confidence intervals were small, thus suggesting that $N_e$ are small rather than large, or noise caused by kinship or rare alleles [63,65]. Other explanations include strong gene flow (which masks genetic drift, resulting in low linkage disequilibrium and therefore high $N_e$ values) [66]. Additionally, violations of the assumptions of the linkage disequilibrium method, for instance closed populations, non-recent bottleneck, or population structure, could also contribute to these results [66]. The genetic parameters $A_r$, $H_o$, $H_e$, $F_{st}$, $P_a$, $F_{is}$ and $N_e$ from translocated and colonized populations were compared against natural populations using R [60] and the packages lme4 (v1.1.26; [67]), lmerTest (v3.1.3; [68]), lmtest (v0.9.38; [69]), and fBasics (v3042.89.1; [70]). The comparisons were done with linear models (genetic parameter ∼ region + population type) using the *lm* function. The factor "region" had two levels (Emmental and Lucerne) and the factor "populations type" had three levels (natural, translocated, and colonized). The model assumptions of normality and independence of residuals were tested for each model using the function *jarqueberaTest* () and *dwtest* (), respectively, from the packages fBasics (v3042.89.1; [70]) and lmtest (v0.9.38; [69]). The tests showed that the assumptions of the linear models were met.

We tested for evidence of genetic bottlenecks using BOTTLENECK (v1.2.02; [71]). We used three models, the stepwise mutation model (SMM) and the two-phase model (TPM) are the most appropriate for testing bottleneck events using microsatellites, and the allele frequency distribution which can differentiate between a bottleneck event and a stable population [72]. The SMM assumes that a mutation happens in a change in one repeat unit. The TPM assumes that a

microsatellites mutation changes between the SMM and that a mutation involves any number of tandem repeats (IAM: Infinite alleles model) [73]. The allele frequency distribution shows the lack of rarest alleles rapidly lost after a bottleneck event [74]. The TPM was run using a recommended parameter set to 95% SMM and 5% IAM, and variance between infinite allele mutations of 12% [72]. Significance was evaluated using a Wilcoxon signed-rank test because of its power and robustness when used with few polymorphic loci [72]. We considered populations bottleneck when two tests, SMM and TPM, were significant because of the uncertainty regarding which mutation model is the most appropriate for micro-satellite loci.

## Genetic population structure and migration rates

To evaluate the genetic effects of translocated populations for the Lucerne sites, the analyses (isolation by distance, genetic pattern as assessed by STRUCTURE and principal component analysis) was conducted for natural and translocated populations together and for natural populations. This was to observe any effect because of the translocations.

The genetic structure of populations from Emmental and Lucerne was analysed using STRUCTURE (v2.3.4; [75–78]). The most probable number of genetic clusters was determined using the highest ΔK values (in STRUCTURE), the observation of the assignment probabilities of each individual for each analysed cluster, and supported by principal component analysis (PCA). The PCA analysis was carried out in the R package adegenet (v2.1.3; [53,54]) using the allele frequencies, centered and scaled, and missing allelic information was substituted with mean values. The results from STRUCTURE were plotted using DISTRUCT (v1.1; [79]).

Markov chain Monte Carlo (MCMC) method using BayesAss (v3.0.4; [80]) were used to estimate the migration rates between populations. The parameters used in the BayesAss analysis for the Emmental populations were the following: random seed = 100, Markov Chain Monte Carlo (MCMC) iterations = 10000000, burn-in = 1000000, sampling interval = 100 and mixing parameters: migration rate (dM) = 0.9, allele frequencies (dA) = 0.9, inbreeding coefficients (dF) = 0.9. For the Lucerne populations the parameter were: random seed = 100, MCMC iterations = 10000000, burn-in = 1000000, sampling interval = 100 and mixing parameters: dM = 0.15, dA = 0.6, dF = 0.6). With this parameterization, the MCMC converged for both populations, Emmental and Lucerne (S3 and S4 Figs, respectively). The obtained migration rates were used in the program quantiNemo ([v2.0.0; [81]) to support the projection of genetic diversity and to assess the future status of the studied populations.

## Projecting genetic diversity

Microsatellite analysis provides information on the current status of the populations. To learn about the future status, we simulated genetic diversity over time to model long-term trends in genetic diversity. To do so, we used the program quantiNemo (v2.0.0; [81]). We used the estimated values of expected heterozygosity ($H_e$) to develop the projections of genetic diversity. $H_e$ is a sensitive genetic parameter to detect reductions in genetic diversity in populations inhabiting disturbed areas [82], to model genetic diversity under different scenarios for 30 generations. We considered a metapopulation selection level, a mean fecundity of 50 with zero fluctuations, a random mating proportion of 0.95 and a promiscuous mating system. Eight different scenarios were simulated for the study areas Emmental and Lucerne. Scenarios accounted for variation in dispersal and population size [83,84]. We assumed different dispersal types, population sizes and dispersal frequencies (for details, see S2 Table). 100 replicates were simulated for each scenario. The results obtained with quantiNemo (v2.0.0; [81]) were analysed with the generalised linear model function *glm* in R [63]: $H_e$ Final ~ $H_e$ Start + Population type (natural vs colonization; natural vs translocated) + Dispersal type + Population size + Dispersal frequency + Population type * Dispersal type + Population type * Population size + Population type * Dispersal frequency. $H_e$ Final and $H_e$ Start refers to the last and first value in the projection of 30 generations, respectively. Separate models were fit to the Emmental and Lucerne data.

## Results

### Genetic diversity and bottlenecks

The genotype data for all 21 and 12 (Emmental and Lucerne, respectively) populations showed negligible frequencies of null alleles (S3 Table). No locus showed consistent deviations from Hardy-Weinberg equilibrium (S3 Table). The few detected deviations from HWE were probably due to population substructure which caused and excess of homozygotes (a Wahlund effect; [85]). Furthermore, there was no strong evidence of linked loci because a low percentage of the variation in one microsatellite marker is shared with the other microsatellite marker (Standardized Index of Association over all loci (rbarD) = 0.019 and 0.062; for Emmental and Lucerne, respectively).

The metrics used to describe genetic diversity in the populations are shown in Fig 2 and Table 1.

The results of the regression models (Table 2) showed no significant effect of region (Emmental and Lucerne) and population type (natural, colonized, and translocated) on the genetic parameters $A_r$, $H_o$, $H_e$, $F_{st}$ and $F_{is}$. This suggests that there were no differences (Fig 2). The genetic parameter $P_a$ and $N_e$ showed a significant effect of region (lower values in Emmental than in Lucerne). $F_{is}$ values in the Emmental colonized populations and in the Lucerne translocated populations were not significantly different from zero (t(8) = −1.075, $p$ = 0.314 and t(5) = −2.251, $p$ = 0.074, respectively), but $F_{is}$ values in the Emmental natural populations and Lucerne natural populations were (t(11) = −3.874, $p$ = 0.003 and t(5) = −3.666 $p$ = 0.015, respectively).

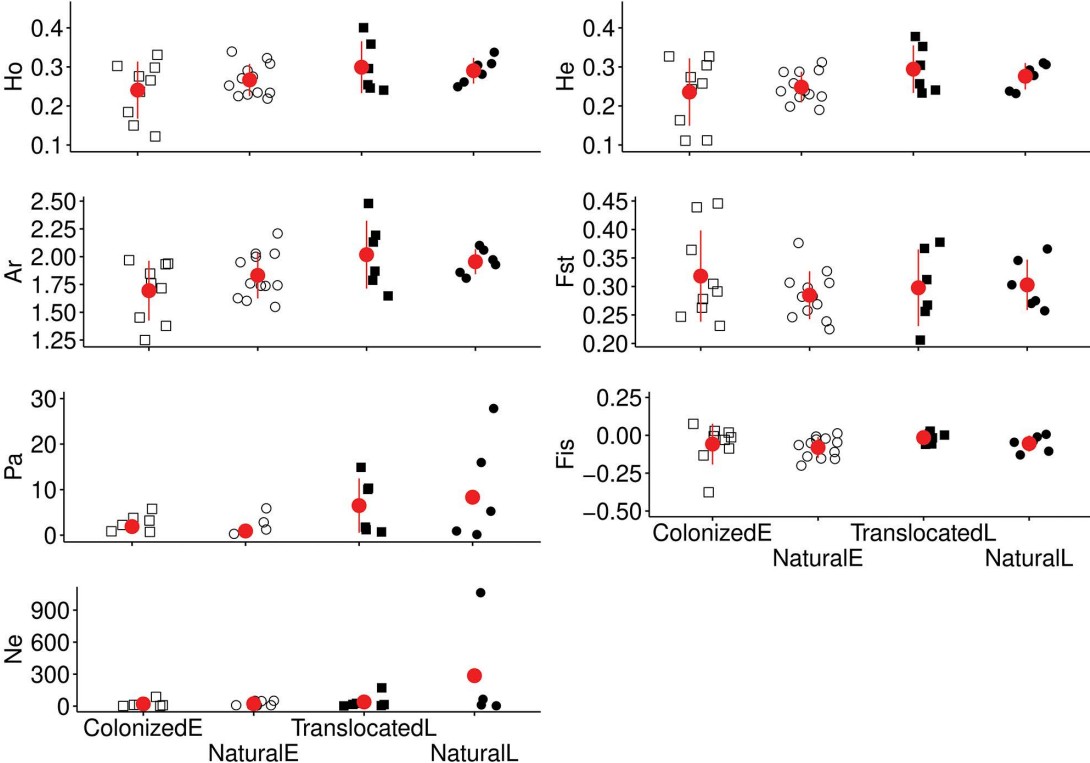

**Fig 2. Strip chart of the genetic diversity indicators for colonized, natural and translocated populations in the study areas Emmental and Lucerne.** $H_o$ = Observed heterozygosity, $H_e$ = Expected heterozygosity, $A_r$ = Allelic richness, $F_{st}$ = Fixation index, $P_a$ = Private alleles, $F_{is}$ = Inbreeding coefficient and $N_e$ = Effective population size. ColonizedE = Emmental colonized populations, NaturalE = Emmental natural populations, TranslocatedL = Lucerne translocated populations and NaturalL = Lucerne natural populations. The figure shows the values for the populations (dots and squares) and the mean values and the standard error (both in red) for the population types. The data presented in this figure is shown in the Table 1. Table 2 shows the statistical analysis of the data.

Table 2. Linear model results for the gene diversity indicators. Allelic richness ($A_R$), observed and expected heterozygosity ($H_o$ and $H_e$, respectively), fixation index ($F_{st}$), private alleles ($P_a$), inbreeding coefficient ($F_{is}$) and effective population size ($N_e$). A significant value is shown by *($P < 0.05$). Values in square brackets are 95% confidence intervals. This analysis shows the absence of significant differences in the genetic diversity indicator between population types (natural, translocated, and colonized). Only $P_a$ and $N_e$ showed significant differences between regions (Emmental and Lucerne).

| | $A_r$ | $H_o$ | $H_e$ | $F_{st}$ | $P_a$ | $F_{is}$ | $N_e$ |
|---|---|---|---|---|---|---|---|
| Intercept | 1.83 | 0.27 | 0.25 | 0.28 | 0.92 | −0.08 | 20.60 |
| | [1.69 − 1.97] | [0.23 − 0.30] | [0.21 − 0.28] | [0.25 − 0.32] | [−2.37–4.20] | [-0.13 − -0.03] | [-149.07 − 190.27] |
| Region: Lucerne | 0.12 | 0.02 | 0.03 | 0.02 | 7.42 | 0.03 | 263.03 |
| | [-0.11 − 0.36] | [-0.03 − 0.08] | [-0.03 − 0.09] | [-0.04 − 0.08] | [1.72 − 13.11] | [-0.06 − 0.11] | [13.27 − 512.78] |
| | p=0.298 | p=0.402 | p=0.353 | p=0.547 | p=0.012* | p=0.551 | p=0.04* |
| Population type: Translocated | 0.06 | 0.01 | 0.02 | −0.01 | −1.83 | 0.04 | −244.38 |
| | [-0.21 − 0.34] | [-0.06 − 0.07] | [-0.05 − 0.09] | [-0.08 − 0.07] | [-8.41 − 4.74] | [-0.06 − 0.14] | [-590.72 − 101.97] |
| | p=0.641 | p=0.784 | p=0.597 | p=0.882 | p=0.573 | p=0.441 | p=0.15 |
| Population type: Colonized | −0.14 | −0.03 | −0.01 | 0.03 | 0.97 | 0.02 | 2 |
| | [-0.35 − 0.07] | [-0.08 − 0.02] | [-0.07 − 0.04] | [-0.02 − 0.09] | [-4.05 − 5.99] | [-0.06 − 0.10] | [−217.05–221.05 |
| | p=0.192 | p=0.293 | p=0.625 | p=0.212 | p=0.695 | p=0.561 | p=0.985 |
| $R^2$ | 0.225 | 0.148 | 0.134 | 0.054 | 0.249 | 0.073 | 0.213 |

The bottleneck analysis found evidence for recent bottlenecks in three populations resulting from colonizations from Emmental (DBM, FEM, and TFM; S4 Table). This revealed significant heterozygosity excess (Wilcoxon test, $p < 0.05$) and the shifted mode test showed a distortion in the allele frequency distribution. The loss of rare alleles and excess of heterozygosity are consistent with a bottleneck occurring within the las few generations. In one of the natural populations in Lucerne (SSS) there was too heterozygosity excess (Wilcoxon test, $p < 0.05$).

## Population structure

STRUCTURE inferred that the Emmental populations were divided into K=2 (ΔK=129.24; S5 Fig). However, based on the hierarchical comparison Ks, and the PC analysis (S5 and S6 Figs; respectively), we concluded that the Emmental population is more likely divided into K=4 clusters (Fig 3). Additionally, the natural and colonized populations in the Emmental are admixed and no population is isolated.

STRUCTURE divided the Lucerne populations (natural and translocated) into K=2 (ΔK=384.42; S7 Fig). However, the hierarchical comparison of Ks and the PC analysis (S7 and S8 Figs; respectively) led us to conclude that the most probable number of clusters was K=3 (Fig 3). STRUCTURE divided the Lucerne natural populations into K=4 (ΔK=5.94; S9 Fig). However, the hierarchical comparison of Ks and the PC analysis (S9 and S10 Figs; respectively) led us to conclude that the Lucerne natural populations are divided into K=3. Both separated analysis (natural vs translocated and natural populations) showed us that the number of clusters in the Lucerne sampled area was three. We showed that natural and translocated populations were admixed. However, the natural population LATL remains as a single cluster. This population was not used as donor population in the translocation plan (S11 Fig).

## Projecting genetic diversity

The results on the simulation of genetic diversity ($H_e$) over time for the Emmental population are presented in Table 3 and S12 Fig.

The projection of genetic diversity for the Emmental study sites showed that there were significant differences in $H_e$ between the genetic diversity of natural and colonized populations at the end of the simulated 30 generations. Average final $H_e$ was 0.236 for natural populations and 0.290 for colonized populations. Plots which visualize the

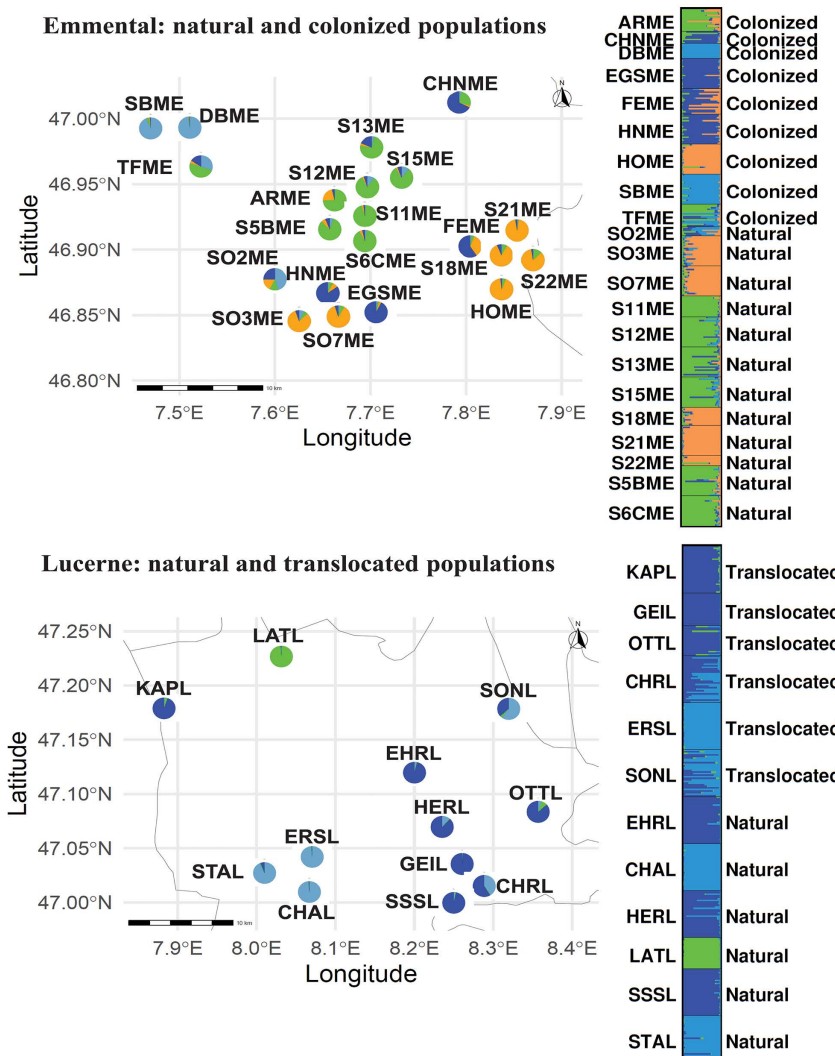

**Fig 3. Pie charts (in map) and STRUCTURE plot showing the cluster assignments for Emmental populations (top; natural and colonized populations) and Lucerne population (bottom; natural and translocated populations).** The STRUCTURE plot for Emmental represents K = 4 (top) and for Lucerne K = 3 (bottom). Each vertical bar in the STRUCTURE plot represents an individual, and the colours within each bar indicate the proportion of that individual's genome assigned to each genetic cluster **(K)**. The number of colours corresponds to the number of clusters (K) inferred in the analysis. Individuals with a single colour are strongly assigned to one cluster, whereas individuals with multiple colours show admixture among clusters. The black lines separate different populations.

interactions between the eight different scenarios are shown in S13 Fig. The number of dispersers had no effect on $H_e$ in colonized populations. In contrast, natural populations had higher $H_e$ with the BayesAss dispersal rates than with five dispersers. Dispersal frequency affected $H_e$ in colonized populations but had no effect in natural populations. In colonized populations, constant dispersal led to higher $H_e$ than dispersal only in the first generation. Population size had no effect on $H_e$ in colonized populations. In contrast, larger natural populations (500) had higher $H_e$ than the smaller natural populations (100).

In the projected genetic diversity for the Lucerne populations, natural and translocated, we found that there were no statistically significant differences in the final genetic diversity values between natural (0.317) and translocated

**Table 3. Generalized linear model results of the genetic diversity ($H_e$) obtained from the simulations in quantiNemo for the Emmental populations.** This analysis considered a metapopulation selection level, with a mean fecundity value of 50 with zero fluctuations, 0.95 of random mating proportion and promiscuity mating system. $R^2 = 0.471$. The values in bold are statistically significant ($P < 0.05$).

| Predictors | $H_e$ Final | | |
| --- | --- | --- | --- |
| | Estimates | 95% CI | p |
| (Intercept) | 0.25 | 0.22–0.28 | **<0.001** |
| $H_e$ Start | 0.35 | 0.23–0.46 | **<0.001** |
| Population type (Natural) | −0.12 | −0.15 – −0.09 | **0.001** |
| Dispersal type (BAYESSAss) | −0.00 | −0.02–0.02 | 0.866 |
| Population size (100) | −0.01 | −0.03–0.01 | 0.392 |
| Dispersal frequency (First generation) | −0.06 | −0.09 – −0.04 | **<0.001** |
| population type (Natural) * Dispersal type (BAYESSAss) | 0.11 | 0.09–0.14 | **<0.001** |
| Population type (Natural) * Population size (100) | −0.05 | −0.08 – −0.02 | **<0.001** |
| Population type (Natural) * Dispersal frequency (First generation) | 0.06 | 0.04–0.09 | **<0.001** |

populations (0.337) (Table 4; S14 Fig). Population size and dispersal frequency affected final $H_e$, i.e., smaller population showed lower $H_e$ and regular dispersal increased $H_e$. There were no significant interactions.

## Discussion

Conservation practice should focus more on genetic diversity [36]. Translocations are a good example where this can be done. Translocations involve the intentional movement of individuals to a new place with the aim to establish a new population. When a new population is founded, it may be genetically less diverse than the population from which the translocated individuals originated. Reduced levels of genetic diversity in translocated populations have been reported in a few cases [23–25] and may compromise the long-term viability of the new population. Reassuringly, our results show that natural, translocated, and populations resulting from natural colonizations had a similar genetic make-up. This is an important result, because if translocated or colonized populations had less genetic diversity than natural populations, then they could be more susceptible to external stressors such as climate change, pathogens or pollution [35,86].

The populations in Lucerne and Emmental showed a genetic structure of multiple spatial clusters. This inference is based on the STRUCTURE analysis and the PCA. The clustering suggests that gene flow is limited, probably because of IBD and landscape heterogeneity [87–89]. Limited gene flow might increase the effects of genetic drift. That is, deleterious

**Table 4. Generalized linear model results of the genetic diversity obtained from the projections of genetic diversity done in quantiNemo for the natural and translocated populations in Lucerne.** This analysis considered a metapopulation selection level, with a mean fecundity value of 50 with zero fluctuations, 0.95 of random mating proportion and promiscuity mating system. $R^2 = 0.366$. The values in bold are statistically significant ($P < 0.05$).

| Predictors | $H_e$ Final | | |
| --- | --- | --- | --- |
| | Estimates | 95% CI | p |
| (Intercept) | 0.30 | 0.23–0.37 | **<0.001** |
| $H_e$ Start | 0.26 | 0.02–0.50 | **0.034** |
| Population type (Translocated) | 0.01 | −0.03–0.05 | 0.606 |
| Dispersal type (BAYESSAss) | 0.00 | −0.03–0.03 | 0.849 |
| Population size (100) | −0.03 | −0.06 – −0.00 | **0.030** |
| Dispersal frequency (First generation) | −0.07 | −0.10 – −0.04 | **<0.001** |
| Population type (Translocated) * Dispersal type (BAYESSAss) | −0.02 | −0.07–0.02 | 0.255 |
| Population type (Translocated) * Population size (100) | 0.02 | −0.02–0.07 | 0.261 |
| Population type (translocated) * Dispersal frequency (First generation) | 0.01 | −0.03–0.05 | 0.700 |

alleles may rise in frequency and populations may have a reduced ability to respond to environmental conditions that individuals may encounter in the new habitat [32], especially in the colonized and translocated populations. An increase in population connectivity through the construction of new ponds [6,9] might help to reduce spatial genetic structure and may increase population viability because there would be a greater exchange of individuals among populations.

The genetic diversity of colonized and translocated populations was similar to that of natural populations. Even though, the median $F_{is}$ values of the Emmental and Lucerne natural populations are significantly different from zero, they do not suggest the presence of inbreeding but of heterozygous (i.e., outbred) individuals due to the mating of distantly related individuals [90]. This heterogeneity may be a characteristic of colonized and translocated populations without the negative costs of outbreeding [91].

Although the power to detect a bottleneck event is small because little time has elapsed since the translocations and colonizations [92], there was evidence for genetic bottleneck events in some colonized and natural populations. Bottlenecks were found in three populations in the Emmental (populations DBM, TFM and FEM) and in a natural (i.e., old) population in Lucerne (populations SSS). The populations in the Emmental are all populations which colonized newly created ponds. We assume that a small number of dispersers may have led to genetic bottlenecks (founder effect [11,12]). In other words, many alleles from the population from where the immigrants originate are not passed on, rare alleles are lost immediately, and genetic drift is strongest in the first generations; even if the population grows rapidly, this does not restore the loss of alleles [93,94]. It is more challenging to explain the bottleneck in the natural population SSS in the Lucerne area. This population inhabits a well and other bodies of water in the grounds of an eighth-century castle. It is conceivable that the maintenance of the well led to the mortality of larvae and thus to bottlenecks. No translocated populations went through a detectable bottleneck event. This suggests that the translocation program might have found a good number of donor populations, distance between natural and translocated populations, quality of the release habitat and the number of translocated individuals [17,95,96]. Despite the bottleneck events that were detected, the genetic structure of the three population types did not differ much (Fig 2, Table 2). Nevertheless, bottleneck reduces genetic diversity, and such a reduction may affect population viability [97]. It could therefore be interesting to compare the demography and viability of the bottlenecked populations with those that show no signs of bottlenecks [26] and to explore whether some form of genetic rescue would be feasible and necessary.

We used simulations to predict future levels of genetic diversity (He) in the populations. Despite the simplifying assumptions that had to be made in the simulations, we found that populations resulting from the colonization of new ponds had higher average final $H_e$ values than natural populations. A possible explanation for the higher mean final $H_e$ values is that the naturally colonized populations may have been colonized by dispersers from multiple populations [98]. This seems likely because, we found that the Emmental populations are admixed and not isolated. However, interactions between dispersal type, dispersal frequency and population size used in the simulations may also have contributed to these results. This suggests that to positively influence dispersal, conservation management should provide high quality habitat for the species by increasing the number of ponds in the study area and by improving the terrestrial habitat (e.g., dry-stone walls and hangslide), and ponds should be allowed to function for long periods [6,9]. To maintain high dispersal rates, one should focus on the three stages of dispersal: emigration, transience and settlement [99]. Emigration could be improved if ponds are constructed and managed in such a way that they produce large numbers of offspring which may emigrate (e.g., through the removal of non-native fish; [100]). During the transient phase, dispersal can be enhanced through the removal of potential barriers to dispersal or the creation of microhabitats which may serve as daytime shelters on potential dispersal corridors [101,102]. Similarly, settlement can be improved by providing suitable habitat. In the case of the midwife toads, microhabitats such as dry-stone walls are important [6]. By doing so, dispersers from source populations would reach other sites in higher numbers and more frequently. All this would prevent genetic diversity from decreasing.

We found that translocations and natural colonizations resulted in populations that were genetically similar to natural populations. Thus, the conservation programmes achieved their primary goal of increasing the number of populations in

the study area. Although maintaining genetic diversity was not an explicit aim of the translocations and pond construction, both programmes achieved this main goal through different approaches. This suggests that both natural colonizations following pond construction and translocations can be used to achieve conservation goals, as long as the risks of translocations are considered (e.g., outbreeding depression, pathogen introduction and adaptive potential; [21,103,104]). However, the evidence of genetic bottlenecks suggests that the translocations can be improved. Because *A. obstetricans* does not disperse over long distances [6,105], the creation of additional ponds could help to increase the number of dispersers and the effective populations size ($N_e$) by expanding breeding opportunities, reducing density-dependent larval mortality, lowering variance in reproductive success, enhancing connectivity among populations, and reducing the impact of environmental stochasticity on recruitment [106,107], and thereby further improve the viability of the population network, both genetically and demographically. Further assessments of the conservation actions should include monitoring of adaptive genetic variation and life history traits related to population viability to detect outbreeding depression and other processes that may reduce population viability in the future [103,104,108,109].

## Supporting information

**S1 Fig. Percentage of missing data for Emmental populations by Locus (on x-axis) and population (on y-axis).** The lowercase letters at the end of the population names indicates the population type (n = natural and c = colonized). Emmental region shows 0.8% missing alleles.
(EPS)

**S2 Fig. Percentage of missing data for Lucerne populations by Locus (on x-axis) and population (on y-axis).** The lowercase letters at the end of the population names indicates the population type (n = natural and t = translocated). Lucerne region shows 0.78% missing alleles.
(EPS)

**S3 Fig. Trace plot of Markov Chain Monte Carlo (MCMC) for inference of migration rate estimates in BayesAss for the Emmental populations.** The grey part of the chain indicates the samples discarded before the migration rate estimates were obtained. The MCMC converged (the trace fluctuates around a stable mean), meaning that the chain ran long enough for the obtained migration rates to be valid and stable.
(EPS)

**S4 Fig. Trace plot of Markov Chain Monte Carlo (MCMC) for inference of migration rate estimates in BayesAss for the Lucerne populations.** The grey part of the chain indicates the samples discarded before the migration rate estimates were obtained. The MCMC converged (the trace fluctuates around a stable mean), meaning that the chain ran long enough for the obtained migration rates to be valid and stable.
(EPS)

**S5 Fig. Bayesian clustering results of the STRUCTURE and delta K plot (in the corner) analysis for microsatellite data of populations sampled in Emmental (natural and colonized populations) of *A. obstetricans* (344 individuals).** The STRUCTURE results show the distribution of clusters from K = 2 to K = 8 using no a priori information on geographic or population. The most suitable K based on the higher delta K (ΔK) is K = 2 (observe ΔK plot too). However, considering the ΔK value, the hierarchical observation of the different clusters and the PCA, K = 4 is best supported. Each vertical bar represents an individual, and the colours within each bar indicate the proportion of that individual's genome assigned to each genetic cluster. The number of colours corresponds to the number of clusters (K) inferred in the analysis. Individuals with a single colour are strongly assigned to one cluster, whereas individuals with multiple colours show admixture among clusters. The black lines separate different populations.
(EPS)

**S6 Fig. Principal components analysis (PCA) for the Emmental, natural and colonized, populations based on 344 individuals and 16 microsatellite loci.** The n and the c at the end of the population's labels mean natural and colonized, respectively. Each point represents an individual, with colours indicating populations. The PCA reveals clustering into 4 groups. The bold bars in the inset graph (eigenvalues) indicate the first two principal components shown in the PCA figure.
(EPS)

**S7 Fig. Bayesian clustering results of the STRUCTURE and delta K plot (in the corner) analysis for microsatellite data of populations sampled in Lucerne (natural and translocated populations) of *A. obstetricans* (330 individuals).** The STRUCTURE results show the distribution of clusters from K=2 to K=10 using no a priori information on geographic or population. The most suitable K based on the higher delta K ($\Delta K$) is K=2 (observe $\Delta K$ plot too). However, considering the $\Delta K$ value, the hierarchical observation of the different clusters and the PCA, K=3 is best supported. Each vertical bar represents an individual, and the colours within each bar indicate the proportion of that individual's genome assigned to each genetic cluster. The number of colours corresponds to the number of clusters (K) inferred in the analysis. Individuals with a single colour are strongly assigned to one cluster, whereas individuals with multiple colours show admixture among clusters. The black lines separate different populations.
(EPS)

**S8 Fig. Principal components analysis (PCA) for the Lucerne, natural and translocated, populations based on 330 individuals and 16 microsatellite loci.** The n and the t at the end of the population's labels mean natural and translocated, respectively. Each point represents an individual, with colours indicating populations. The PCA reveals clustering into 3 groups. The bold bars in the inset graph (eigenvalues) indicate the first two principal components shown in the PCA figure.
(EPS)

**S9 Fig. Bayesian clustering results of the STRUCTURE and delta K plot (in the corner) analysis for microsatellite data of populations sampled in Lucerne (natural populations) of *A. obstetricans* (170 individuals).** The STRUCTURE results show the distribution of clusters from K=2 to K=6 using no a priori information on geographic or population. The most suitable K based on the higher delta K ($\Delta K$) is K=4 (observe $\Delta K$ plot too). However, considering the $\Delta K$ value, the hierarchical observation of the different clusters and the PCA, K=3 is best supported. Each vertical bar represents an individual, and the colours within each bar indicate the proportion of that individual's genome assigned to each genetic cluster. The number of colours corresponds to the number of clusters (K) inferred in the analysis. Individuals with a single colour are strongly assigned to one cluster, whereas individuals with multiple colours show admixture among clusters. The black lines separate different populations.
(EPS)

**S10 Fig. Principal components analysis (PCA) for the Lucerne natural populations based on 170 individuals and 16 microsatellite loci.** The n at the end of the population's labels means natural. Each point represents an individual, with colours indicating populations. The PCA reveals clustering into 3 groups. The bold bars in the inset graph (eigenvalues) indicate the first two principal components shown in the PCA figure.
(EPS)

**S11 Fig. Circos plot showing the translocations.** Number of individual translocated from donor populations are represented by the arrows from left (donor/natural) to right (translocated). Each arrow represents a translocation event. The size of the axis shows the total of individuals translocated. The letters n and t at the end of the names represent natural and translocated populations, respectively.
(EPS)

**S12 Fig. Projecting 30 generations of genetic diversity in the Emmental populations under eight scenarios considering a metapopulation selection level, with a mean fecundity value of 50 with zero fluctuations, 0.95 of random mating proportion and promiscuity mating system.** 1=ARME colonized, 2=CHNM colonized, 3=DBME colonized, 4=EGSME colonized, 5=FEME colonized, 6=HNME colonized, 7=HOME colonized, 8=SBME colonized, 9=TFME colonized, 10 = SO2ME natural, 11 = SO3ME natural, 12 = SO7ME natural, 13=S11ME natural, 14=S12ME natural, 15=S13ME natural, 16=S15ME natural, 17=S18ME natural, 18=S21ME natural, 19=S22ME natural, 20=S5BME natural and 21=S6CME natural. Lines correspond to the heterozygosity respond to the migration rate estimates in BayesAss and the eight simulated scenarios explained in Table S2. The variation over time in $H_e$ values among Emmental population types (natural and colonized) is more sensitive to changes in dispersal type, population size, and dispersal frequency (see Table 3 for details).
(EPS)

**S13 Fig. Final expected heterozygosity ($H_e$) for colonized and natural Emmental populations under various simulated scenarios (dispersal type, population size, and dispersal frequency).** Points represent mean $H_e$ (obtained over 30 generations), with error bars showing standard error. Colonized populations consistently exhibit higher $H_e$ across scenarios compared to natural populations. Thus, demographic assumptions strongly influence the magnitude of $H_e$ in natural populations.
(EPS)

**S14 Fig. Projecting 30 generations of genetic diversity in the Lucerne populations (translocated and natural) under eight scenarios considering a metapopulation selection level, with a mean fecundity value of 50 with zero fluctuations, 0.95 of random mating proportion and promiscuity mating system.** 1=KAPL translocated, 2=GEIL translocated, 3=OTTL translocated, 4=CHRL translocated, 5=ERSL translocated, 6=SONL translocated, 7=EHRL natural, 8=CHAL natural, 9=HERL natural, 10=LATL natural, 11=SSSL natural and 12=STAL natural. Lines correspond to the heterozygosity respond to the migration rate estimates in BayesAss and the eight simulated scenarios explained in Table S2. In general, smaller populations showed lower heterozygosity ($H_e$) and regular dispersal increased $H_e$. The variation over time in $H_e$ values does not show variation between Lucerne, translocated and natural populations (see Table 4 for details).
(EPS)

**S1 Table. New microsatellite DNA markers of Alytes obstetricans developed by ecogenics GmbH.** Repeat type is based on genomic DNA sequence analyzed on an Illumina MiSeq platform. Size bp (base pairs) is based on the fragment analysis of 15 individuals on an ABI373.
(DOCX)

**S2 Table. Details for the simulations made in quantiNemo (v2.0.0) for Emmental and Lucerne populations.** This table shows eight simulated scenarios for dispersal frequency, population size, and dispersal type.
(DOCX)

**S3 Table. Estimated deviation from Hardy-Weinberg equilibrium (HWE) and null allele frequencies per population and for the 16 microsatellite loci.** A) Shows the p-values of the HWE exact test, Monte Carlo permutations of alleles. Significant deviation from HWE is shown by *($P<0.05$). B) Shows the estimates of null allele frequencies. None of the loci showed consistent deviation from Hardy-Weinberg equilibrium, and null allele frequencies were negligible.
(DOCX)

**S4 Table. Results on the Bottleneck analysis on natural, colonized and translocated populations from canton Lucerne and Emmental using three different calculation models.** The * represents statistically significant results

($p < 0.05$). L-shaped distributions indicate a stable population with numerous low-frequency alleles, whereas mode-shifted distributions reflect a recent bottleneck, characterized by the loss of rare alleles and a shift toward intermediate allele frequencies.
(DOCX)

## Acknowledgments

We thank all pond owners and conservation authorities in Emmental and Lucerne for granting us access, allowing us sampling the endangered amphibians, and sharing information on the populations. For field assistance we thank Sam Cruickshank and Ursina Tobler, and Chris Funk for comments on the manuscript. Our appreciation to the group of Prof. Lukas Keller, especially to Glauco Camenisch and Martina Schenkel for their help during the lab work. Last but not least, we would like to thank all the people whose conservation efforts improved the conservation status of the midwife toad and thereby made this study possible.

## Author contributions

**Conceptualization:** José F. Meléndez-Cal-y-Mayor, Benedikt R. Schmidt.

**Data curation:** José F. Meléndez-Cal-y-Mayor, Jasmin Winkler, Ramon Müller.

**Formal analysis:** José F. Meléndez-Cal-y-Mayor.

**Funding acquisition:** José F. Meléndez-Cal-y-Mayor, Benedikt R. Schmidt.

**Investigation:** José F. Meléndez-Cal-y-Mayor, Jasmin Winkler, Ramon Müller, Benedikt R. Schmidt.

**Methodology:** José F. Meléndez-Cal-y-Mayor, Jasmin Winkler, Ramon Müller, Benedikt R. Schmidt.

**Project administration:** Benedikt R. Schmidt.

**Resources:** Benedikt R. Schmidt.

**Supervision:** Arpat Ozgul, Benedikt R. Schmidt.

**Validation:** José F. Meléndez-Cal-y-Mayor, Benedikt R. Schmidt.

**Visualization:** José F. Meléndez-Cal-y-Mayor, Benedikt R. Schmidt.

**Writing – original draft:** José F. Meléndez-Cal-y-Mayor, Benedikt R. Schmidt.

**Writing – review & editing:** José F. Meléndez-Cal-y-Mayor, Jasmin Winkler, Ramon Müller, Beatrice Lüscher, Janine Bolliger, Arpat Ozgul, Benedikt R. Schmidt.

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
