## [Decision Letter · Decision Letter 0]

2 Jul 2025

Dear Dr. Meléndez,

Thank you for submitting your manuscript to PLOS ONE. After careful consideration, we feel that it has merit but does not fully meet PLOS ONE’s publication criteria as it currently stands. Therefore, we invite you to submit a revised version of the manuscript that addresses the points raised during the review process.

Reviewer 1 made some valuable suggestions, especially regarding the statistical analyses, which should be carefully considered during the revision. Please also make sure that the figures are displayed as intended. In the current version, many of the figures are distorted or incomplete and therefore were not easily reviewed. 

We look forward to receiving your revised manuscript.

Kind regards,

Sven Winter

Academic Editor

PLOS ONE

Journal Requirements:

“This project was made possible through the funds of the University of Zurich, Department of Evolutionary Biology and Environmental Studies and the Consejo Nacional de Humanidades Ciencias y Tecnologías, Mexico (CONAHCYT).”

“This project was made possible through the funds of the University of Zurich, Department of Evolutionary Biology and Environmental Studies and the Consejo Nacional de Humanidades Ciencias y Tecnologías, Mexico (CONAHCYT).”

Reviewers' comments:

Reviewer's Responses to Questions

**Comments to the Author**

1. Is the manuscript technically sound, and do the data support the conclusions?

Reviewer #1: Yes

2. Has the statistical analysis been performed appropriately and rigorously?

Reviewer #1: I Don't Know

3. Have the authors made all data underlying the findings in their manuscript fully available?

Reviewer #1: Yes

4. Is the manuscript presented in an intelligible fashion and written in standard English?

Reviewer #1: Yes

Reviewer #1: This paper provides a valuable contribution to understanding the genomic patterns of naturally and artificially colonized ponds, and compares them to original source populations. I appreciate the authors' use of existing opportunities to assess the genomic status of an organism of conservation concern, a commendable approach to monitoring these populations. The manuscript employs a range of population genetic tools and statistical approaches to assess and project genetic diversity across population types. Overall, the methods appear thoughtfully selected and generally appropriate for the study's objectives. I offer the following comments and suggestions to help improve clarity and robustness:

Introduction:

The statement, “Beatrice Lüscher made sure that no translocations took place in these naturally colonised populations,” is mentioned. However, without additional context regarding who Beatrice Lüscher is outside the author section and how they ensured no translocations, it is difficult to fully understand the significance of this statement.

Sampling:

The study uses a good narrow sampling timeframe, which is especially relevant for a species that reproduces annually. This strengthens the validity of their results.

One point of clarification: I am unsure if the tadpoles sampled survived the sampling process. If the organisms do not survive, it would be helpful to specify this to avoid any potential confusion.

Methods:

The laboratory methods used are sound, and bioinformatics techniques for assessing linkage disequilibrium (LD), Hardy-Weinberg equilibrium (HWE), null alleles, and percent missing data appear appropriate and well-executed. For genetic diversity and bottleneck detection, AR and private alleles are strongly associated with sample size. Larger sample sizes typically contain more alleles compared to smaller samples. It seems in your results that this pattern doesn't obscure your results, which I think makes them even more meaningful. The authors use mean Fis to assess whether it significantly differs from zero. However, Fis values are often not normally distributed across microsatellites. Given this, I suggest using the median Fis to summarize the data, as it may offer a more robust estimate, or demonstrate normality.

Statistical Analyses:

The use of linear models to compare genetic diversity indicators across regions and population types is valid for continuous response variables. However, I recommend clarifying whether the model assumptions (e.g., normality) were checked, particularly for metrics like Fst and Fis. These metrics may require transformations before fitting a linear model, and this should be addressed in the methods section.

On line 160, the authors suggest that the Ne values reaching infinity could be due to sampling error. I encourage the authors to review the literature for other possible explanations of this phenomenon, as it may not be solely a result of sampling error.

The use of BOTTLENECK is appropriate for detecting recent bottlenecks, and the authors selected models suited to microsatellite data. However, I recommend reporting the specific significant p-values, rather than just using the asterisk notation, to allow for clearer interpretation. Additionally, given the weight placed on bottlenecks in the discussion, the results section could benefit from more detailed reporting of the bottleneck results, including data from the supplemental section.

The use of quantiNemo to simulate genetic diversity loss under different demographic and dispersal scenarios is a strong approach for understanding future genetic diversity trajectories. However, I recommend clarifying whether multiple replicates were simulated for each scenario, as stochasticity plays a significant role in forward-time simulations.

Figures:

I was unable to view Figure 1 properly, and I expect the version in the PDF is distorted compared to what was submitted. However, even considering that, the figure is tricky to interpret. I only see orange dots on the main map, and I wonder if the purple dots are covered. If the final version of the figure does not contain purple dots, this should be addressed. In addition, the inset map could benefit from using the same color-coding as the main map. To make the figure more accessible at a glance, I recommend changing the shape of the dots for different site types, in addition to using lowercase letters for differentiation.

Supplementary Information:

I believe Supplemental Table 1 provides important information that would benefit from being moved to the main paper rather than remaining in the supplements. Its content appears central to the study and would improve the overall readability of the paper if included in the main manuscript.

Supplemental Table 4 is difficult to read due to formatting issues. Despite attempts to adjust the table in my Word document, I was unable to view all the columns at once. I recommend improving the formatting for better accessibility.

Overall, this paper presents a compelling study on genetic diversity patterns in a conservation context, and the methodology is generally sound. A few clarifications and additions would strengthen the manuscript, particularly in terms of statistical analysis and presentation of results. The authors use of current genetic diversity data and forward simulations is a notable strength, and I believe these findings will make a valuable contribution to the field.

**Do you want your identity to be public for this peer review?** For information about this choice, including consent withdrawal, please see our Privacy Policy

Reviewer #1: No

---

## [Author Response · Author response to Decision Letter 1]

5 Sep 2025

Dear Editor,

In this document we explain how we have revised the manuscript in response to the comments made by the editor and the reviewers. Our responses start with RESPONSE >. Line numbers correspond to the files with track changes.

The reviews were very constructive and helped to improve the manuscript.

Best wishes,

on behalf of all authors

PONE-D-25-16547

Translocated populations are genetically similar to natural populations and populations resulting from natural colonizations

PLOS ONE

Dear Dr. Meléndez,

Thank you for submitting your manuscript to PLOS ONE. After careful consideration, we feel that it has merit but does not fully meet PLOS ONE’s publication criteria as it currently stands. Therefore, we invite you to submit a revised version of the manuscript that addresses the points raised during the review process.

RESPONSE > Thank you very much for your comments. We are happy to continue improving this paper. Below, we describe all of the changes we have made in response to your comments and those of the reviewer.

Reviewer 1 made some valuable suggestions, especially regarding the statistical analyses, which should be carefully considered during the revision. Please also make sure that the figures are displayed as intended. In the current version, many of the figures are distorted or incomplete and therefore were not easily reviewed.

RESPONSE > Thank you for the comments. We explain below how we revised the manuscript in response to the comments. Here we would just like to point out that we reran the statistical analysis with Fis as suggested and that we fixed the problem with the figures.

Journal Requirements:

RESPONSE > We have made the required changes following the suggested templates to fulfil the journal requirements.

RESPONSE > We added the request information to the manuscript (from line 97 to 99 and from 132 to 137). Please note that we had an animal welfare permit to conduct the study. In this permit, all animal welfare issues are described and specified.

RESPONSE > The data used for this study is already available on Zenodo with DOI: https://doi.org/10.5281/zenodo.15052694

“This project was made possible through the funds of the University of Zurich, Department of Evolutionary Biology and Environmental Studies and the Consejo Nacional de Humanidades Ciencias y Tecnologías, Mexico (CONAHCYT).”

RESPONSE > We have removed the funding information from the Acknowledgements Section and from the manuscript.

“This project was made possible through the funds of the University of Zurich, Department of Evolutionary Biology and Environmental Studies and the Consejo Nacional de Humanidades Ciencias y Tecnologías, Mexico (CONAHCYT).”

RESPONSE > The Funding Statement we have provided is correct. We don’t wish to do any amendments.

RESPONSE > We made a new map using data which is all free to use. Specifically, we used map data which only uses publicly available data. The data is available in the R package “rnaturalearth” and “rnaturalearthdata”.

RESPONSE > We made a new map for which we do not need a permit.

RESPONSE > Thank you for the suggested resources. However, we used a different package in R for which we do not need a permit.

Reviewers' comments:

Reviewer's Responses to Questions

Comments to the Author

1. Is the manuscript technically sound, and do the data support the conclusions?

Reviewer #1: Yes

2. Has the statistical analysis been performed appropriately and rigorously?

Reviewer #1: I Don't Know

3. Have the authors made all data underlying the findings in their manuscript fully available?

Reviewer #1: Yes

4. Is the manuscript presented in an intelligible fashion and written in standard English?

Reviewer #1: Yes

5. Review Comments to the Author

Reviewer #1: This paper provides a valuable contribution to understanding the genomic patterns of naturally and artificially colonized ponds, and compares them to original source populations. I appreciate the authors' use of existing opportunities to assess the genomic status of an organism of conservation concern, a commendable approach to monitoring these populations. The manuscript employs a range of population genetic tools and statistical approaches to assess and project genetic diversity across population types. Overall, the methods appear thoughtfully selected and generally appropriate for the study's objectives. I offer the following comments and suggestions to help improve clarity and robustness:

RESPONSE > Thank you for your positive assessment of our manuscript.

Introduction:

The statement, “Beatrice Lüscher made sure that no translocations took place in these naturally colonised populations,” is mentioned. However, without additional context regarding who Beatrice Lüscher is outside the author section and how they ensured no translocations, it is difficult to fully understand the significance of this statement.

RESPONSE > We added the information to the manuscript that Beatrice Lüscher is the local amphibian conservation officer.

Sampling:

The study uses a good narrow sampling timeframe, which is especially relevant for a species that reproduces annually. This strengthens the validity of their results.

One point of clarification: I am unsure if the tadpoles sampled survived the sampling process. If the organisms do not survive, it would be helpful to specify this to avoid any potential confusion.

RESPONSE > It is a threatened species. Therefore, there was no lethal sampling. This is now explained in the section “Research permits and ethical consideration”.

Methods:

The laboratory methods used are sound, and bioinformatics techniques for assessing linkage disequilibrium (LD), Hardy-Weinberg equilibrium (HWE), null alleles, and percent missing data appear appropriate and well-executed. For genetic diversity and bottleneck detection, AR and private alleles are strongly associated with sample size. Larger sample sizes typically contain more alleles compared to smaller samples. It seems in your results that this pattern doesn't obscure your results, which I think makes them even more meaningful. The authors use mean Fis to assess whether it significantly differs from zero. However, Fis values are often not normally distributed across microsatellites. Given this, I suggest using the median Fis to summarize the data, as it may offer a more robust estimate, or demonstrate normality.

RESPONSE > We used the median Fis values as suggested by the reviewer to assess whether it significantly differs from zero. Only the result for Lucerne natural populations changed from no significant to significantly different from zero (line 157; from line 259 to 264; line 362; Table S1).

Statistical Analyses:

The use of linear models to compare genetic diversity indicators across regions and population types is valid for continuous response variables. However, I recommend clarifying whether the model assumptions (e.g., normality) were checked, particularly for metrics like Fst and Fis. These metrics may require transformations before fitting a linear model, and this should be addressed in the methods section.

RESPONSE > The model assumption of normality of residuals was checked for each model. For all models, the null hypothesis of normality was not rejected (From line 172 to 175).

On line 160, the authors suggest that the Ne values reaching infinity could be due to sampling error. I encourage the authors to review the literature for other possible explanations of this phenomenon, as it may not be solely a result of sampling error.

RESPONSE > We have added two possible more explanations in accord to our findings (from line 162 to 166).

The use of BOTTLENECK is appropriate for detecting recent bottlenecks, and the authors selected models suited to microsatellite data. However, I recommend reporting the specific significant p-values, rather than just using the asterisk notation, to allow for clearer interpretation. Additionally, given the weight placed on bottlenecks in the discussion, the results section could benefit from more detailed reporting of the bottleneck results, including data from the supplemental section.

RESPONSE > We have added the p-values in Table S4 for the models SMM and TPM. For the allele frequency distribution there is not p-values but frequencies across loci and tends to follow an L-shaped distribution a shifted mode. After a recent bottleneck rare alleles are lost more rapidly than intermediate-frequency alleles. This shifts the allele frequency distribution. This is known as shifted mode. So, we can only report if the frequencies of alleles across loci have a L-shape distribution or a shifted mode (presence of bottleneck). This all is explained in the references we cited in the manuscript

---

## [Decision Letter · Decision Letter 1]

22 Oct 2025

Dear Dr. Meléndez-Cal-y-Mayor,

We look forward to receiving your revised manuscript.

Kind regards,

Sven Winter

Academic Editor

PLOS ONE

Journal Requirements:

Reviewers' comments:

Reviewer's Responses to Questions

**Comments to the Author**

Reviewer #2: All comments have been addressed

2. Is the manuscript technically sound, and do the data support the conclusions?

Reviewer #2: Yes

3. Has the statistical analysis been performed appropriately and rigorously?

Reviewer #2: Yes

4. Have the authors made all data underlying the findings in their manuscript fully available?

Reviewer #2: Yes

5. Is the manuscript presented in an intelligible fashion and written in standard English?

Reviewer #2: Yes

Reviewer #2: I have carefully examined the authors’ point-by-point responses to the previous review as well as the revised version of the manuscript. Overall, it is clear that the authors have made a substantial effort to address the earlier concerns, and many of the revisions have improved the clarity, methodological transparency, and overall quality of the paper. In particular, the handling of the statistical comments appears sound: the use of median Fis values was implemented as suggested, model assumptions were tested and reported, and the treatment of bottleneck results has been improved. Ethical aspects related to animal handling have been clarified with commendable precision, and the data availability now meets the journal’s requirements. These are all meaningful improvements.

The manuscript itself presents a well-defined and relevant study that addresses the important question of genetic similarity of translocated, colonized, and natural populations. The research is well framed and generally methodologically solid. The use of multiple analytical approaches and the integration of empirical data with forward simulations are notable strengths that make this study potentially valuable to the conservation genetics community.

However, some of the responses remain rather superficial. This is especially true for the interpretation of infinite Ne values, which is acknowledged but not convincingly discussed, and for the section on bottleneck events, where additional depth and contextualization would strengthen the argument (please also check all references again, some are missing, e.g., l. 180 "residuals were tested for each model using the function jarqueberaTest () and dwtest ()"). The adjustments to figures and tables still fall short of making the results fully accessible at a glance. For example, the revised map is likely clearer than the original, but the description suggests that the overall visualization strategy remains basic rather than genuinely improved, and the whole map (still?) appears broken to me. Please check your visualization tool. Moreover, all figures and tables need a proper caption, which is self-explanatory; e.g., table 1 lacks a conclusive caption.

A general pattern throughout the responses is that several replies address the comments formally, but without fully exploiting the opportunity to clarify or deepen the presentation of the results. For example, the discussion of the infinite Ne values just lists two additional explanations without critically evaluating their likelihood in the specific study context or providing supporting literature. The response to the bottleneck-related comment adds some details but remains descriptive and lacks an interpretation of how these results inform the study’s main conclusions. Likewise, the comment on improving figure clarity was addressed by creating a new map, but without any clear rationale for how the new visualization strategy better communicates the underlying patterns. While the core analyses seem sound and the technical points were largely addressed, the interpretation and communication of the findings could still be more thorough.

In sum, the authors have clearly engaged with the reviewers’ feedback and improved the manuscript in several important respects. Yet some areas would benefit from more thoughtful revision, particularly regarding interpretation of key genetic results (biological context) and data presentation. I would therefore recommend minor revisions, with a focus on strengthening the clarity and depth of interpretation as well as improving the accessibility of figures and tables.

**Do you want your identity to be public for this peer review?** For information about this choice, including consent withdrawal, please see our Privacy Policy

Reviewer #2: No

---

## [Author Response · Author response to Decision Letter 2]

17 Dec 2025

Dear Editor,

Thank you for giving us another opportunity to revise our manuscript. In this document we explain how we have revised the manuscript in response to the comments made by the editor and the reviewers. Our responses start with RESPONSE >. Line numbers correspond to the files with track changes.

We deleted text from the previous decision letter that is not relevant here.

The reviews were very constructive and helped to improve the manuscript.

Best wishes,

on behalf of all authors

Subject: PLOS ONE Decision: Revision required [PONE-D-25-16547R1]

PONE-D-25-16547R1

Translocated populations are genetically similar to natural populations and populations resulting from natural colonizations

PLOS ONE

Dear Dr. Meléndez-Cal-y-Mayor,

Thank you for submitting your manuscript to PLOS ONE. After careful consideration, we feel that it has merit but does not fully meet PLOS ONE’s publication criteria as it currently stands. Therefore, we invite you to submit a revised version of the manuscript that addresses the points raised during the review process.

I agree with the new reviewer that the manuscript did greatly improve and that many of the previous comment have been addressed but there are still a few comment that have not been answered thoroughly enough.

RESPONSE > Thank you. Below we explain how we improved the manuscript. In the last revision, we gave the most consideration to those comments that served to improve the answers to our scientific questions.

We look forward to receiving your revised manuscript.

Kind regards,

Sven Winter

Academic Editor

PLOS ONE

Journal Requirements:

RESPONSE > We have checked our reference list, and they are complete and correct.

6. Review Comments to the Author

Reviewer #2: I have carefully examined the authors’ point-by-point responses to the previous review as well as the revised version of the manuscript. Overall, it is clear that the authors have made a substantial effort to address the earlier concerns, and many of the revisions have improved the clarity, methodological transparency, and overall quality of the paper. In particular, the handling of the statistical comments appears sound: the use of median Fis values was implemented as suggested, model assumptions were tested and reported, and the treatment of bottleneck results has been improved. Ethical aspects related to animal handling have been clarified with commendable precision, and the data availability now meets the journal’s requirements. These are all meaningful improvements.

The manuscript itself presents a well-defined and relevant study that addresses the important question of genetic similarity of translocated, colonized, and natural populations. The research is well framed and generally methodologically solid. The use of multiple analytical approaches and the integration of empirical data with forward simulations are notable strengths that make this study potentially valuable to the conservation genetics community.

RESPONSE > Thank you for the positive assessment of the manuscript and the revision.

However, some of the responses remain rather superficial. This is especially true for the interpretation of infinite Ne values, which is acknowledged but not convincingly discussed,

RESPONSE > We added more specific explanations of the potential reasons why we found infinite Ne values based on previous works and on our data (from line 160 to 165). We explain that we think that infinite Ne are most likely an artefact.

and for the section on bottleneck events, where additional depth and contextualization would strengthen the argument

RESPONSE > We added more additional information to strengthen our argument (from line 372 to 385). We explain why we believe that there were bottlenecks in the recently colonized populations (simply because colonization caused the bottleneck). In one of the “old” natural populations in Lucerne area, we believe it was due to maintenance of the well.

(please also check all references again, some are missing, e.g., l. 180 "residuals were tested for each model using the function jarqueberaTest () and dwtest ()").

RESPONSE > The citation brackets in PlosOne are []. The brackets that the reviewer mentioned as lacking a citation correspond to the functions jarqueberaTest and dwtest (it is R language). These brackets indicate that both are functions, and inside the brackets, the parameters are specified. The functions jarqueberaTest and dwtest do not require a citation because they are included in the fBasics package, which is already cited (lines 176-177).

The adjustments to figures and tables still fall short of making the results fully accessible at a glance. For example, the revised map is likely clearer than the original, but the description suggests that the overall visualization strategy remains basic rather than genuinely improved, and the whole map (still?) appears broken to me. Please check your visualization tool.

RESPONSE > It is a simple map which shows the location of the adjacent study areas and the populations. We used different colours for the two study areas. We believe that the map serves its purpose. We double-checked all files during the submission process to make sure that the maps are displayed correctly.

Moreover, all figures and tables need a proper caption, which is self-explanatory; e.g., table 1 lacks a conclusive caption.

RESPONSE > We have included more details in the captions of all figures and tables.

A general pattern throughout the responses is that several replies address the comments formally, but without fully exploiting the opportunity to clarify or deepen the presentation of the results. For example, the discussion of the infinite Ne values just lists two additional explanations without critically evaluating their likelihood in the specific study context or providing supporting literature.

RESPONSE > We added some additional explanations for Ne and bottlenecks. We don’t think that a further discussion is useful because it would be speculative. Regarding the infinite Ne values, we added additional explanations and supporting literature in the manuscript (from line 160 to 165). This new information explains the possible reasons why the Ne value was infinite for each population.

The response to the bottleneck-related comment adds some details but remains descriptive and lacks an interpretation of how these results inform the study’s main conclusions.

RESPONSE > We added in the manuscript more interpretation on the presence of bottleneck in 3 colonized populations and in a natural population (from line 374 to 385). This connects with the discussion we do on the presence of bottleneck from line 424 to 430. The main conclusion is that the three population types do not differ much despite the bottlenecks. We added some text in the discussion which clarifies this.

Likewise, the comment on improving figure clarity was addressed by creating a new map, but without any clear rationale for how the new visualization strategy better communicates the underlying patterns.

RESPONSE > We believe that the new maps convey that necessary information. Fig 1 shows the two study areas and all the populations while Fig 3 shows that spatial genetic structure. The maps are simple, but all the necessary information is there. There is no additional information in the maps that may distract (e.g. Hill shade).

While the core analyses seem sound and the technical points were largely addressed, the interpretation and communication of the findings could still be more thorough.

RESPONSE > We added additional explanations in the discussion in response to the previous comments and hope that this made the interpretation and discussion of the results more thorough.

In sum, the authors have clearly engaged with the reviewers’ feedback and improved the manuscript in several important respects.

RESPONSE > Thank you.

Yet some areas would benefit from more thoughtful revision, particularly regarding interpretation of key genetic results (biological context) and data presentation. I would therefore recommend minor revisions, with a focus on strengthening the clarity and depth of interpretation as well as improving the accessibility of figures and tables.

RESPONSE > We explain in the replies to the previous comments how we improved the manuscript.

---

## [Editor Report · Decision Letter 2]

23 Dec 2025

Translocated populations are genetically similar to natural populations and populations resulting from natural colonizations

PONE-D-25-16547R2

Dear Dr. Meléndez-Cal-y-Mayor,

We’re pleased to inform you that your manuscript has been judged scientifically suitable for publication and will be formally accepted for publication once it meets all outstanding technical requirements.

Kind regards,

Sven Winter

Academic Editor

PLOS One
---

## [Editor Report · Acceptance letter]

PONE-D-25-16547R2

PLOS One

Dear Dr. Meléndez-Cal-y-Mayor,

I'm pleased to inform you that your manuscript has been deemed suitable for publication in PLOS One. Congratulations! Your manuscript is now being handed over to our production team.

Kind regards,

on behalf of

Dr. Sven Winter

Academic Editor

PLOS One